# Effects of the Morphology, Surface Modification and Application Methods of ZnO-NPs on the Growth and Biomass of Tomato Plants

**DOI:** 10.3390/molecules25061282

**Published:** 2020-03-12

**Authors:** Eneida A. Pérez Velasco, Rebeca Betancourt Galindo, Luis A. Valdez Aguilar, José A. González Fuentes, Bertha A. Puente Urbina, Samuel A. Lozano Morales, Saúl Sánchez Valdés

**Affiliations:** 1Departamento de Horticultura, Universidad Autónoma Agraria Antonio Narro, Saltillo C.P. 25315, Coah, Mexico; velasco.ady@gmail.com (E.A.P.V.); luisalonso_va@hotmail.com (L.A.V.A.); jagf@uaaan.edu.mx (J.A.G.F.); 2Departamento de Materiales Avanzados, Centro de Investigación en Química Aplicada, Saltillo C.P. 25294, Coah, Mexico; bertha.puente@ciqa.edu.mx; 3Departamento de Síntesis de Polímeros, Centro de Investigación en Química Aplicada, Saltillo C.P. 25294, Coah, Mexico; alejandro.lozano@ciqa.edu.mx; 4Departamento de Procesos de Transformación de Plásticos, Centro de investigación en Química Aplicada, Saltillo C.P. 25294, Coah, Mexico; saul.sanchez@ciqa.edu.mx

**Keywords:** greenhouse crops, surface modification, maltodextrin, plant biomass

## Abstract

Benefits of nanotechnology in agriculture include reduced fertilizer loss, improved seed germination rate and increased crops quality and yield. The objective of this research was to evaluate the effects of zinc oxide nanoparticles (ZnO-NPs), at 1500 ppm, on tomato (*Solanum lycopersicum* L.) growth. ZnO-NPs were synthetized to produce either spherical or hexagonal morphologies. In this research, we also studied two application methods (foliar and drench) and nanoparticles’ (NPs) surface modification with maltodextrin. The results obtained indicate that ZnO-NP-treated tomato plants significantly increased plant height, stem diameter and plant organs (leaves, stem and root) dry weight compared to plants without NP treatment.

## 1. Introduction

Nanotechnology (NT) is a multidisciplinary science that has gained importance in agriculture and other economic activities including industries such as the textile, cosmetics, medicine, electronics, food production and hydraulics industries [1,2,3,4,5,6,7]. Nanotechnology in agriculture has attained acceptance in the last decade because it has arisen as a technological advancement to produce different tools; for example, nanoproducts can be used as nanosensors for stress detection caused by biotic and abiotic factors, nanopesticides to increase pesticide efficiency [8] and nanofertilizers (Nf). Nanofertilizers have been defined as modified fertilizers that may improve crop productivity and quality, as well as soil fertility [9]. It is known that agriculture faces different challenges that threaten food production sustainability: climatic change, population increase and the extensive use of chemicals that pollute soil, water and plants, causing ecological imbalance and directly affecting animals and humans [10]. The particular interest in Nf is to reduce the rate of chemical fertilizers required to achieve increased crop growth and yield [11].

Zinc (Zn) plays a key role in enzymatic activation for protein synthesis in plants. It is considered an essential microelement because it is required in small quantities but it is also crucial for vegetative development [12]. This nutrient acts as a precursor in phytohormones like auxins, which influence cell elongation and division, furthermore, Zn is essential for photosynthesis and facilitates carbohydrate metabolism in plants because Zn stabilizes or activates the proteins involved in these processes [13].

Zinc, along with iron, copper, silver and titanium, is among the metals most commonly used for the synthesis of nanoparticles (NPs). Numerous studies have reported that plant development and growth respond to ZnO-NP application and that they have the capacity to inhibit and control some diseases [11].

Tarafdar mentioned that ZnO-NPs increase the activity of phytase and alkaline and acid phosphatases, contributing to phosphorus solubilization and plant uptake [14]. The ZnO-NPs are also reported to (1) increase by 21.6% the number of shoots in chickpea seedlings [15], (2) increase by 10% the germination of cucumber seeds compared to the control (Zn) [16], (3) stimulate flowering up to 14 days earlier in onions [17], and (4) increase the length of the main and lateral roots by 17% and 26%, respectively, in tobacco plants treated with ZnO-NPs compared to plants treated with ZnSO_4_ [18]. The application of ZnO-NPs to *Setaria italica* increased seed oil and nitrogen content and plants exhibited higher water stress tolerance [19]. There are many benefits from the use of ZnO-NPs; nevertheless, it should be noted that they present a photo-catalytic activity that gives rise to oxidative reactions in the particle’s surface, unleashing free radicals that promote degradation ([20].

However, ZnO-NPs exhibit stability and dispersion problems due to their nanometric size. NPs often gather together causing particle agglomeration [21] due to their wide surface area, therefore presenting poor dispersion [22]. In order to decrease NPs’ agglomeration and oxidation, and to improve their dispersion and stability, scientists have investigated different methods of NP coating or surface modification. Several works on ZnO-NPs’ modification or coating with aggregated organic and inorganic compounds and polymeric matrixes have been reported [23,24,25,26]. ZnO-NP modification with different types of modifiers is summarized in Figure 1.

Maltodextrin (MDX) is a polysaccharide derived from starch hydrolysis and its main application is in the food industry as an artificial sweetener [27]. It is also used in agriculture as a constituent in some insecticides and acaricides [28]. It has been shown that NPs’ morphology affects their optical, electrochemical, sensory, thermal and mechanical properties. This morphology effect is a phenomenon known as magnetic anisotropy [29]. Particle morphology also contributes to the dispersion, degradation process, stability and compatibility of ZnO-NPs [30]. However, despite the previous investigations conducted in this field, the ZnO-NPs’ morphology effect, as well as the effect of surface modification, have not been thoroughly explored.

Mexico is one of the world’s main tomato (*Solanum lycopersicum* L.) producers and exporter [31]. Tomato nutritional value is high and is appreciated due to its vitamins, minerals, sugars, proteins, fiber, organic acids and lycopene, an antioxidant with anticarcinogenic qualities [32]. The objective of this research was to determine the effect of ZnO-NPs with two different morphologies (spherical or hexagonal) in interaction with MDX modification and two application methods (foliar or drench), on the growth parameters of tomato plants.

## 2. Materials and Methods

### 2.1. NPs Synthesis

Two ZnO-NPs morphologies (spherical and hexagonal) were prepared using 26.33 g of Zn (O_2_CCH_3_)_2_, 1700 mL of ethanol and 300 mL of deionized water. Immediately, 5.36 mL of triethanolamine (TEA) and 1.42 mL of n-propylamine (NPA) were added. The reaction was conducted at 65 °C in reflux and with constant stirring for 6 h (for hexagonal NPs synthesis) or 12 h (for spherical NPs synthesis). After the system was cooled, the material obtained was rinsed, centrifuged and dried under a vacuum for 12 h at 80 °C. The precipitation method described by Hsieh [33] was used in this research.

### 2.2. ZnO-NPs’ Surface Modification

The ZnO-NPs’ surface modification was conducted using an MDX:ZnO-NPs 1:1 molar ratio (1.5 g:1.5 g). Ethanol was used as a dispersion agent. The reaction was carried out at 65 °C in reflux and with constant stirring for 6 h. The system was immediately cooled. The modified ZnO-NPs were decanted and dried out for 12 h at 80 °C for their further analysis.

### 2.3. ZnO-NPs Characterization

The infrared analysis (FT-IR) was carried out using Nicolet iS50 equipment (Thermo Fisher Scientific Inc., Madison, WI, USA.) KBr tablets were prepared with ZnO-NPs samples to identify the ZnO-NPs and the respective modifying agent. The XRD analysis was conducted with a Siemens D-500 diffractometer (SIEMENS, Munich, GER) with CuKα radiation to identify the crystalline phase in both modified and non-modified ZnO-NPs. The morphology and surface modification of ZnO-NP samples were carried out through a high-resolution transmission electronic microscope (HRTEM) Titan 80–300 kV (FEI Company, Hillsboro, OR, USA).

### 2.4. Plant Material and Management

The effect of ZnO-NPs was measured at experiment termination on tomato cv Clermon plants grown under greenhouse conditions. Seeding was carried out on 13 February, 2018, using 1 L containers filled with sphagnum peat.

Two NP application methods were used: drench and foliar. Drench was conducted before planting with 50 mL of a 1500 ppm ZnO-NPs solution dispersed by a sonicator (SONICS model VC505) for 15 min at 38% Abs. Foliar application was conducted manually five weeks after planting, when leaves were fully developed. Plants were transplanted nine weeks after seeding in 10 L containers with a mixture of sphagnum peat and perlite (60%:40% *v*/*v*). A second ZnO-NP application was performed five weeks after transplanting, both via drench or foliar.

Plants were fed with a Steiner’s [34] nutrient solution adjusted to the plant’s phenological stage through a drip irrigation system with two 4 L/h emitters. Lateral shoots and older foliage were pruned weekly. The experiment was terminated on August 2018.

### 2.5. Growth and Plant Organs Biomass Measurement

Plant height and stem diameter were measured at experiment termination, while pruned leaves were collected and placed on a drying oven at 65 °C for 48 h prior recording the dry weight. Root and stem dry weight was measured at experiment termination and after dried in an oven, as previously described, for leaves.

### 2.6. Statistical Data Analyses

Treatments were arranged in a randomized block design using a factorial arrangement with 12 treatments and 5 replications, one plant per replication. The factors considered were: two ZnO-NPs morphologies (spherical and hexagonal, plus a control with no NPs application), two application methods (drench and foliar) and NPs surface modification with MDX (modified and non-modified). The software used for the statistical analysis was SAS (SAS Version 9.4, SAS Institute, Cary, NC, USA) and a means comparison test was conducted according to Tukey’s multiple comparison test (*p* < 0.05).

## 3. Results and Discussion

### 3.1. ZnO-NPs Characterization

The HRTEM micrograph of the ZnO-NPs reveals the presence of spherical particles of <30 nm and with an even distribution size (Figure 2a). The larger particles are attributed to particle agglomeration. The particle size distribution histogram indicates that it ranges from 11 to 40 nm, with an average of 22.5 nm (Figure 2b). Additionally, hexagonal NPs (Figure 3a–b) with single particles ranging from 60 to 120 nm and an average of 85 nm were obtained. ZnO-NPs’ morphology, size and distribution observed in our study are similar to those reported by Hsieh [33], who performed the NPs synthesis with the precipitation method.

Figure 4a,b shows MDX-modified ZnO-NPs HRTEM images showing the presence of agglomerates as well as a fairly even MDX coating over the hexagonal and spherical NPs surface; Coating thickness was 0.92 and 1.22 nm for hexagonal and spherical NPs, respectively, suggesting that particle agglomeration does not affect surface modification.

MDX-modified and non-modified NPs infrared spectra are shown in Figure 5. A belt appears at 450 cm^−1^ in both hexagonal and spherical nanoparticle spectrum due to the stretching of the ZnO bond. MDX-modified ZnO-NP spectra verifies that, in both morphologies, NPs were superficially coated with MDX. This was confirmed by MDX absorption bands at 1000 cm^−1^ as well as the band corresponding to ZnO which are present in both spectrums. In this way, FT-IR-aided, it is perceived that both ZnO-NPs synthesis and modification were properly carried out.

In Figure 6, non-modified ZnO-NPs (Figure 6a,b) and MDX-modified ZnO-NPs (Figure 6c) corresponding X-ray (DRX) diffraction patterns are shown. Peaks present in DRX patterns match with wurtzite-type crystalline structures corresponding to the ZnO standard (JCPDS 36-1451) [35,36]. This means that the surface modification at which the NPs were subjected did not affect their crystalline structure [22,24].

### 3.2. ZnO-NPs Effects on Plant Growth and Biomass

Nanoparticle morphology, surface modification with MDX and application method significantly affected tomato growth (Table 1). Stem diameter increased when hexagonal ZnO-NPs were applied, whilst surface modification with MDX and application method did not affect this parameter. However, an interaction between the assessed factors, such as the interaction between morphology and surface modification, and between morphology and application method, was observed (Table 1).

Plant height was higher when hexagonal, MDX-modified NPs where applied, however, the application method did not have a significant effect (Table 1). Every assessed factor interacted among them, having a significant influence on plant height (Table 1). Hexagonal ZnO-NPs, as well as MDX modification, significantly increased leaf dry weight; however, the application method, foliar or drench, did not affect this parameter. Root and stem dry weight displayed a similar response, since they were not affected by ZnO-NPs morphology; however, MDX surface modification as well as drench application increased root and stem dry weight (Table 1).

Stem diameter increased with spherical and hexagonal NPs application, but this occurred only without MDX. In comparison, there was no effect when modified NPs were applied (Figure 7a). Stem diameter increased when spherical and hexagonal NPs via foliar were applied. However, no effect was observed when a drench application was used (Figure 7b).

Plant height increased when MDX-modified hexagonal NPs were applied (Figure 8a) and when in control plants, when MDX but no NPs were applied via drench (Figure 8b), suggesting that the polysaccharide used in NP modification is acting as a plant growth stimulant. However, this effect may be enhanced when used along with ZnO-NPs. MDX has been used as an ingredient for very few agrochemicals like stimulants and has displayed beneficial effects in crop growth and development, as in lettuce, in which fresh biomass and yield increased 69% and 64%, respectively [37].

Leaf dry weight increased with both hexagonal and spherical NPs application, although surface modification had no influence. However, MDX application without using NPs resulted in a substantial leaf dry weight increase compared to the control without MDX (Figure 9a), confirming that MDX has a growth regulation effect. Foliar dry weight increased with hexagonal and spherical NPs application, but it was more substantial when drench application with hexagonal NPs was used (Figure 9b).

MDX without NPs application increased stem dry weight, nevertheless, using MDX and hexagonal NPs exhibited a slight further increase in stem dry weight (Figure 10a). On the other hand, a decrease in stem dry weight with NPs application, regardless of their morphology and application mode, was observed. There was an increase in stem dry weight in plants treated without ZnO-NPs, only with MDX, as long as it was by drench application (Figure 10b).

MDX-modified hexagonal NPs increased root dry weight compared to control plants (Figure 11a). On the other hand, MDX application without ZnO-NPs drench-applied significantly increases root dry weight (Figure 11b). This result is similar to that on the stem dry weight. Syu reported that using spherical NPs in *Arabidopsis* plants and roots increased their growth compared to triangular NPs [38].

The positive effects of ZnO-NPs in plants, as observed in this research, have not been completely understood, however, greater absorption and retention of nutrients by plants has been reported when nanometric materials are applied [39], ZnO-NPs have the capacity of increasing enzyme activity, such as phytase, alkaline and acid phosphatase, which may contribute to nutrient solubilization like phosphorus [14]. Even though the effect of NP morphology is not very well understood yet, it is clear that is of utmost importance to take advantage of NPs’ physical and chemical properties. Morphology, along with other characteristics that NPs should have, such as purity, crystalline state and size, determine the final product’s yield, since NP distribution and ion release rate depend on them [40].

The application of materials at a nanoscale has still unknown interactions with plants, however, it is more clear that the application of ZnO-NPs improves not only growth but also biomass accumulation, as reported in tobacco plants as ZnO positively affected the stem diameter, length and dry root weight [18]. Faizan reported enhanced growth, photosynthetic attributes, increased antioxidant activity and increased protein accumulation in tomato plants exposed to ZnO-NPs on untreated plants [41].

There are no studies in agriculture that prove either NPs’ morphology or MDX coating effects. However, studies in medicine that explain that, as there is no control of the morphology in the synthesis of nanoparticulated systems, pharmaceuticals use surface coating to release the medical drugs in a controlled fashion, to avoid intoxicating consumers [42]. We hypothesize that the effect of surface-modified and MDX-coated ZnO-NPs on tomato plants may be due to this controlled released attribute.

## 4. Conclusion

ZnO-NP was synthesized with spherical and hexagonal morphologies and the procedure for modifying or coating NPs with maltodextrin was established. Tomato plants treated with ZnO-NP significantly improved plant height, stem diameter and dry weights, usually with hexagonal morphology and superficially modified. The modification of ZnO-NPs with MDX enhances the effect of ZnO-NPs, however, the application via drench of MDX without ZnO-NPs, in general, improved plant growth, supporting the hypothesis that MDX acts as a plant stimulant.

## Figures and Tables

**Figure 1 molecules-25-01282-f001:**
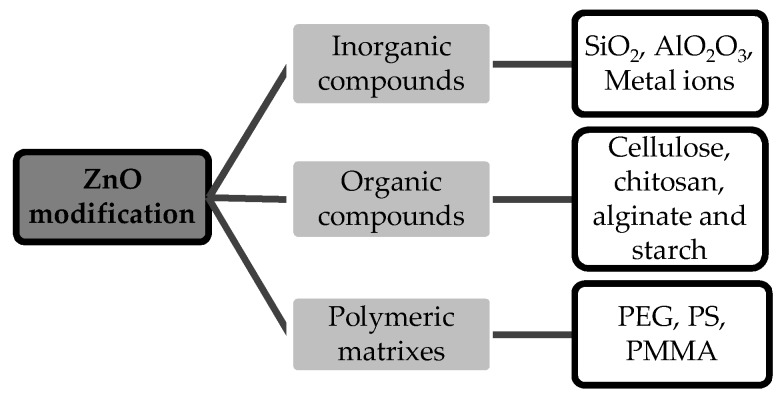
Used agents for surface modification of ZnO. PEG = polyethylene glycol; PS = polystyrene; PMMA = polymethylmethacrylate.

**Figure 2 molecules-25-01282-f002:**
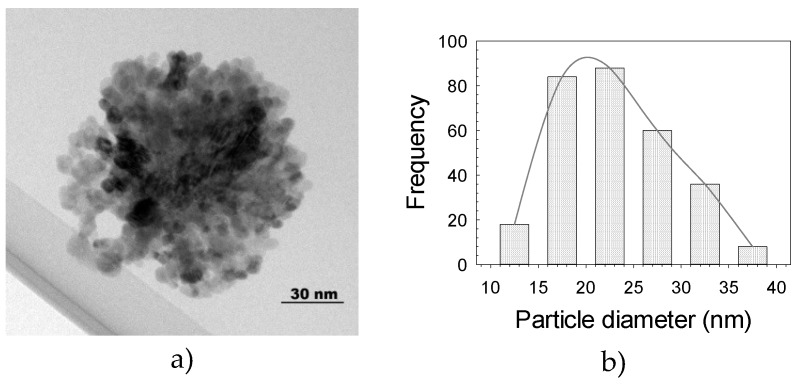
Transmission electron microscopy micrograph showing the spherical morphology of ZnO-NPs (**a**) and histogram of particle size distribution (**b**).

**Figure 3 molecules-25-01282-f003:**
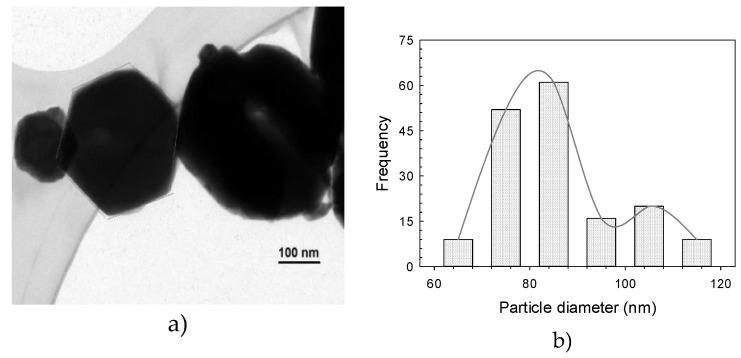
Transmission electron microscopy micrograph showing the hexagonal morphology of ZnO-NPs (**a**), and histogram of particle size distribution (**b**).

**Figure 4 molecules-25-01282-f004:**
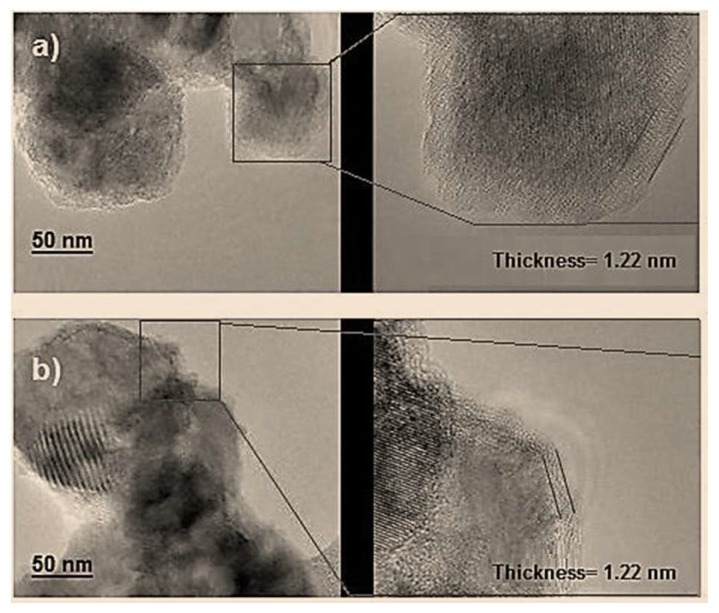
Transmission electron microscopy micrograph. Malotodextrine-modified ZnO-NPs with (**a**) spherical and (**b**) hexagonal morphology.

**Figure 5 molecules-25-01282-f005:**
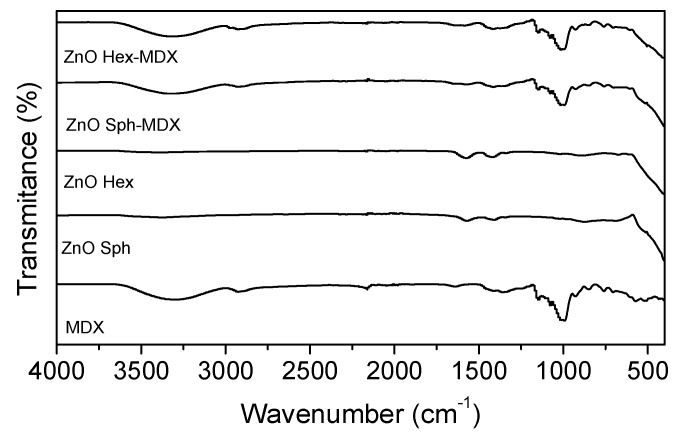
FT-IR spectrums of ZnO-NPs synthesized and modified by precipitation method.

**Figure 6 molecules-25-01282-f006:**
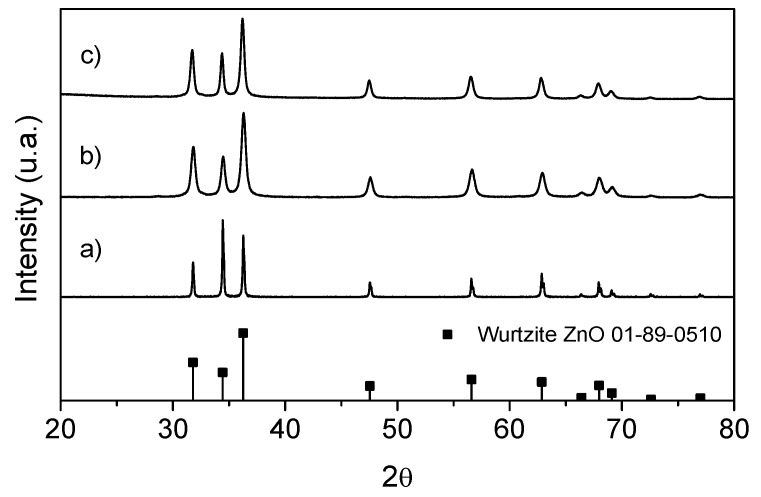
ZnO nanoparticle X-ray diffractogram. (**a**) Spherical ZnO-NP, (**b**) hexagonal ZnO-NP and (**c**) maltodextrin-modified ZnO-NP.

**Figure 7 molecules-25-01282-f007:**
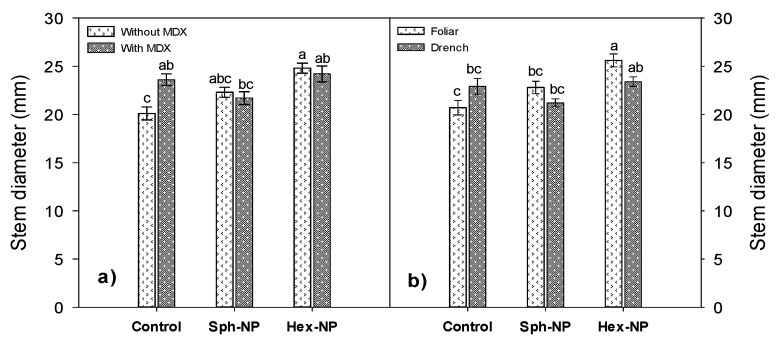
Effect of ZnO-NPs application on stem diameter. (**a**) Morphology*MDX interaction, (**b**) morphology and application method interaction. Bars represent the standard error of the mean. Different letters indicate significant differences according to Tukey’s multiple comparison test (*p* < 0.05).

**Figure 8 molecules-25-01282-f008:**
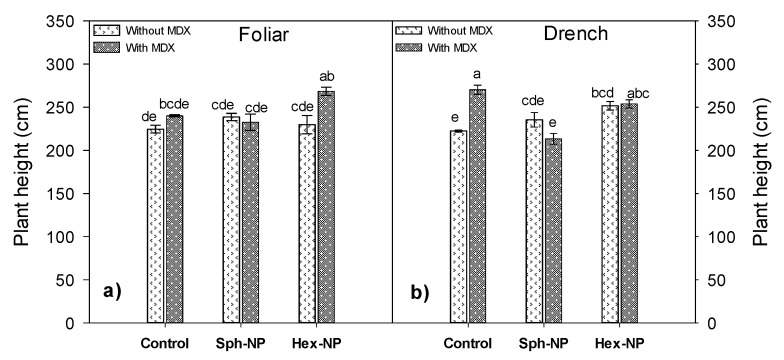
Effect of ZnO-NPs application on plant height. (**a**) Morphology*MDX*foliar application interaction, (**b**) Morphology*MDX*drench application interaction. Bars represent the standard error of the mean. Different letters indicate significant differences according to Tukey’s multiple comparison test (*p* < 0.05). Sph = spherical, Hex = hexagonal.

**Figure 9 molecules-25-01282-f009:**
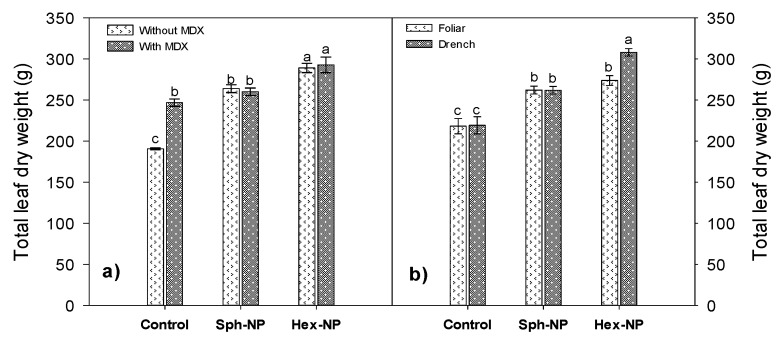
Effect of ZnO-NPs application on leaf dry weight. (**a**) Morphology and maltodextrin (MDX) modification interaction, (**b**) morphology and application method interaction. Bars represent the standard error of the mean. Different letters indicate significant differences according to Tukey’s multiple comparison test (*p* < 0.05).

**Figure 10 molecules-25-01282-f010:**
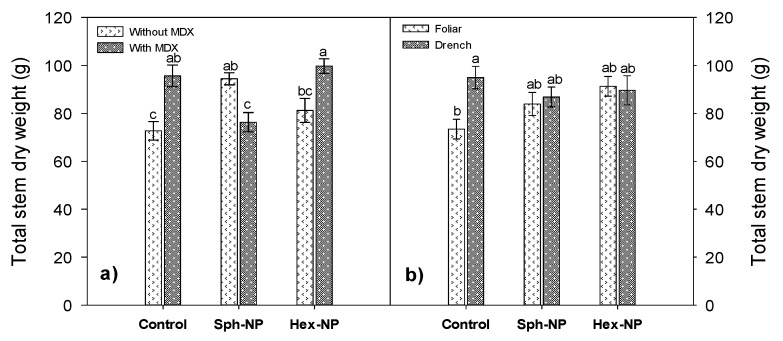
Effect of ZnO-NPs application on stem dry weight. (**a**) Morphology and maltodextin (MDX) interaction, (**b**) morphology and application method interaction. Bars represent the standard error of the mean. Different letters indicate significant differences according to Tukey’s multiple comparison test (*p* < 0.05).

**Figure 11 molecules-25-01282-f011:**
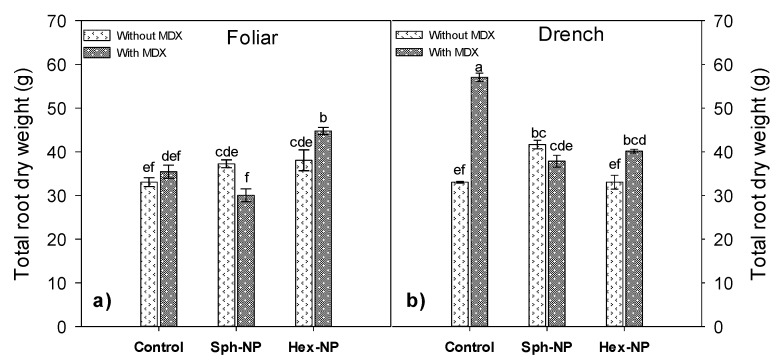
Effect of ZnO-NPs application on root dry weight. (**a**) Morphology*maltodextine (MDX)*foliar application interaction, (**b**) Morphology*MDX*drench application interaction. Bars represent standard error of the mean. Different letters indicate significant differences according to Tukey’s multiple comparison test (*p* < 0.05). Sph = spherical, Hex = hexagonal.

**Table 1 molecules-25-01282-t001:** Effect of different morphology, modification or non-modification with maltodextrine, and application methods of ZnO-NPs on different growth parameters of tomato plants.

	Stem Diameter (mm)	Plant Height (cm)	Leaf Dry Weight (g)	Root Dry Weight (g)	Stem Dry Weight (g)
Morphology					
Control	21.8 b	239 b	223.9 c	39.7 a	84.2 a
Spherical	21.9 b	230 b	253.2 b	36.7 b	85.4 a
Hexagonal	24.5 a	251 a	278.5 a	39.0 a	90.5 a
ANOVA	*p* ≤ <.0001	*p* ≤ <.0001	*p* ≤ <.0001	*p* ≤ 0.0045	*p* ≤ 0.1396
MDX					
Non-Modified	22.4 a	234 b	245.3 b	36.2 b	82.8 b
Modified	23.1 a	246 a	258.4 a	40.9 a	90.5 a
ANOVA	*p* ≤ 0.0898	*p* ≤ 0.0006	*p* ≤ <.0001	*p* ≤ <.0001	*p* ≤ 0.006
Application method					
Foliar	23.0 a	239 a	252.3 a	36.4 b	82.9 b
Drench	22.5 a	241.1 a	251.4 a	40.5 a	90.4 a
ANOVA	*p* ≤ 0.2243	*p* ≤ 0.5336	*p* ≤ 0.0035	*p* ≤ <.0001	*p* ≤ 0.0073
Interactions					
M*MDX	*p* ≤ 0.0003	*p* ≤ <.0001	*p* ≤ <.0001	*p* ≤ <.0001	*p* ≤ <.0001
M*A	*p* ≤ 0.0003	*p* ≤ 0.015	*p* ≤ 0.0006	*p* ≤ <.0001	*p* ≤ 0.0024
MDX*A	*p* ≤ 0.5866	*p* ≤ 0.3323	*p* ≤ 0.7354	*p* ≤ <.0001	*p* ≤ 0.0301
M*MD*A	*p* ≤ 0.2135	*p* ≤ 0.0005	*p* ≤ 0.6123	*p* ≤ <.0001	*p* ≤ 0.6338

M = morphology, MDX = maltodextrin, A = application. Different letters in same column are statistically different according to Tukey’s multiple comparison test, (*p* < 0.05).

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
