# Peer review of "Effects of the Morphology, Surface Modification and Application Methods of ZnO-NPs on the Growth and Biomass of Tomato Plants"

_molecules, 2020, doi:10.3390/molecules25061282_

Round 1

Reviewer 1 Report

In this manuscript by Pérez Velasco et al., the authors describe the effects of ZnO nanoparticles (ZnO-NPs) on the growth of tomato (Solanum lycopersicum) plants. Different ZnO-NPs, with and without maltodextrin surface modification, and different application methods were used. Multiple parameters of plant growth and vitality were investigated. Statistical data analysis using Tukey’s test identified several parameters that were significantly affected by Zn-NP treatment.

These studies complement previous work by several laboratories on ZnO-NPs and their effects on plant growth, including their effects on tomatoes. They have done a good job of characterizing their ZnO-NPs and determining those parameters most affected. Generally, their conclusions are reasonable from the data presented. However, I have a few questions that, if addressed, could improve their manuscript for eventual publication.

  1. There has been a number of prior publications on ZnO-NPs and the growth of different plants, including tomatoes. A partial list might include G. de la Rosa et al. (2013) Pure Appl. Chem. 85, 2161-2174; SV Raskar & SL Laware (2014) Int. J. Curr. Microbiol. Appl. Sci. 3, 467-473; SL Laware & S. Raskar (2014) Int. J. Curr. Microbiol. Appl. Sci. 3, 874-881; MT Maryam et al. (2019) Funct. Plant Biol. 46, 360-375; M Kolenčík et al. (2019) Nanomaterials 9, 1559; XP Wang et al. (2018) Biol Plant 62, 801–808; M Faizan et al. (2019) J. Plant Biochem. Biotechnol. s13562-019-00525-z. I think it would be an improvement to cite these in the Introduction and perhaps compare and contrast their findings with those of others in their Discussion.
  2. [Materials and Methods 2.2] While a 1:1 molar ratio is indicated for MDX:ZnO-NP synthesis, more useful would be a description of the actual concentrations used.
  3. [Results and Discussion] I would suggest that this section be divided into two parts: 3.1 ZnO-NP characterization, and 3.2 ZnO-NP effects on tomato growth.
  4. [Figures] Some of the text within Figures 2b, 3b, 5, and 6 were difficult to read. This should be corrected.
  5. [Figure 6] What is the lowest trace? Wurtzite-type ZnO standard spectra? Please indicate in the figure legend.
  6. [Table 1] As it is presently formatted, it is extremely difficult to appreciate the statistical findings shown in this table. I do not believe this is the preferred way to present such data and its Tukey’s statistical analysis. Perhaps separating the original data from the statistical analysis as two tables would be better?
  7. [Figures 7-11] While these bar graphs are a more familiar means of presenting their data, they would be enhanced by the addition of pairwise bracket and asterisk indications of statistical significance, as typically seen in prior publications.
  8. [Conclusions] In many of their experiments, the application of maltodextrin without ZnO-NPs had a pronounced effect on tomato growth. While indicated in the individual studies, this observation should be restated in the Conclusion.

Author Response

Response to Reviewer 1

We appreciate the time it takes to evaluate the article we submit to this journal. Thank you for your outstanding collaboration and valuable contributions.

Reviewer 2 Report

In this paper, Eneida A. Pérez Velasco et al. synthetized ZnO-NPs with both spherical and hexagonal morphologies, and the NP maltodextrin coating was established NPs-ZnO treated tomato plants significantly improved the growth of plant, including plant height, stem diameter and dry weights. This paper is interesting and useful. There's still some minor revision that i suggest for it's publication as seen below.

  1. The texts and caption in the Figures 2,3,4 can not see clearly. And the corresponding annotations in the Figures should be enlarged.
  2. The scale bar should be given in the Fig. 4.
  3. For the studies of the application and multidisciplinary science of nanotechnology in the first paragraph of Introduction, the authors can refer these paper:1. Nanoscale, 2020, 12, 4077-4084 Nanoscale, 2019,11, 17607-17614 .
  4. The language should be improved.

Author Response

(The authors gave the same response as above.)
